# Fusion MRI/Ultrasound-Guided Transperineal Biopsy: A Game Changer in Prostate Cancer Diagnosis

**DOI:** 10.3390/jcm14020453

**Published:** 2025-01-12

**Authors:** Ilias Giannakodimos, Aris Kaltsas, Napoleon Moulavasilis, Zisis Kratiras, Dionysios Mitropoulos, Michael Chrisofos, Konstantinos Stravodimos, Evangelos Fragkiadis

**Affiliations:** 1Third Department of Urology, Attikon University Hospital, School of Medicine, National and Kapodistrian University of Athens, 12462 Athens, Greece; ares-kaltsas@hotmail.com (A.K.); zkratiras@gmail.com (Z.K.); mxchris@yahoo.com (M.C.); 2Department of Urology, Laikon General Hospital, National and Kapodistrian University of Athens, 11527 Athens, Greece; napomoul@hotmail.com (N.M.); dmitrop@med.uoa.gr (D.M.); kgstravod@yahoo.com (K.S.); e.fragkiadis@gmail.com (E.F.)

**Keywords:** prostate cancer diagnosis, transperineal biopsy, clinically significant prostate cancer, targeted biopsy, systematic prostate biopsy, prostate cancer imaging

## Abstract

**Background/Objectives**: Multiparametric-Magnetic Resonance Imaging(mp-MRI) presents the ability to detect clinically significant cancer, aiming to avoid biopsy if the results are negative or target an abnormal lesion if a suspected lesion of the prostate is found. Recent guidelines recommend the performance of 12 standard biopsies along with 3 to 5 targeted biopsies in suspected prostate lesions, depending on the size of the prostate lesion. In addition, prostate biopsy can be performed by either the transperineal or the transrectal approach. The aim of this comprehensive review is to highlight the role of both standard and targeted MRI/Ultrasound (US) fusion transperineal biopsy (TPB) in the diagnostic approach of prostate cancer cases, to report its diagnostic efficacy and complication rates and to suggest the promising usage of MRI/US fusion TPB in the future. **Methods:** A comprehensive review of the existing literature, including systematic reviews, meta-analyses, and clinical guidelines, was conducted to compare the efficacy and safety of transperineal and transrectal approaches in prostate cancer detection. Special emphasis was placed on mp-MRI-guided targeted biopsy and its combination with systematic sampling. **Results:** Prostate biopsy via the transperineal approach is related to increased detection rates, especially for anterior lesions, and decreased infection risk compared to the transrectal approach, while complication rates (hematuria, hemospermia, etc.) remain similar. Due to lower infection rates via the transperineal route, the performance of prostate biopsy using the transperineal approach is strongly recommended. Finally, transperineal fusion MRI/US biopsy can be valuable for repeat biopsies in patients who had an initial negative biopsy or for the follow-up of patients that undergo active surveillance. **Conclusions:** MRI/US fusion-guided TPB represents a significant advancement in prostate cancer diagnostics, combining improved precision with reduced infection risks. Although TPB presents higher detection rates for anterior prostatic lesions and lower post-biopsy infection rates, there is no significant difference in cancer detection rates compared to TRB. Targeted training and investment may reduce long-term expenses of TPB by lowering hospitalizations, antibiotic usage, and related costs. Future research should further refine this approach and explore its integration with emerging technologies like artificial intelligence for enhanced lesion targeting and diagnostic accuracy.

## 1. Introduction

Prostate cancer remains a significant public health challenge, as it is one of the most commonly diagnosed malignancies in men worldwide. Early and accurate diagnosis is pivotal for optimizing clinical outcomes and minimizing the risks of overtreatment [1]. Typically, men presenting with elevated prostate-specific antigen (PSA) levels or abnormal findings on digital rectal examination (DRE) undergo an evaluation via standard transrectal ultrasound (TRUS)-guided biopsy. However, this method has inherent limitations, including the reduced detection of clinically significant prostate cancer (csPCa) and a tendency to identify clinically insignificant tumors [2,3].

Recent advancements in imaging technologies, particularly the adoption of multiparametric magnetic resonance imaging (mp-MRI), have revolutionized the diagnostic approach. Mp-MRI enables the precise localization of suspicious lesions and improves csPCa detection rates while reducing unnecessary biopsies [4]. Emerging techniques, such as MRI/ultrasound fusion-guided biopsies and the use of high-resolution micro-ultrasound, have further improved diagnostic accuracy [5].

Additionally, the TPB approach has gained prominence due to its lower risk of infectious complications compared to the transrectal route. When combined with local anesthetic protocols, this technique has facilitated effective outpatient procedures with high diagnostic efficacy and minimal morbidity [6]. Moreover, the integration of novel biomarkers and liquid biopsy techniques is shaping a new paradigm for patient selection and personalized cancer diagnostics. The combination of advanced imaging and molecular diagnostics holds promise for refining active surveillance strategies and mitigating overtreatment [7].

In this evolving landscape, the role of multiparametric and targeted biopsy methods is expanding, offering improved diagnostic precision and patient outcomes. This review explores recent advancements in prostate biopsy techniques, emphasizing their clinical implications and future perspectives.

## 2. Methods

A comprehensive literature search was conducted through January 2024 to identify relevant articles. The PubMed database served as the primary source, and only English-language publications were included. The following search terms were used: “Transperineal Fusion MRI/US prostate biopsy”, “Fusion MRI/US prostate biopsy”, and “Transperineal prostate biopsy”. The searches focused on prostate biopsy techniques, MRI/US fusion biopsies, perilesional biopsy, complications, and safety considerations. Systematic reviews, meta-analyses, and clinical guidelines that compared the efficacy and safety of transperineal and transrectal approaches in prostate cancer detection were selected. Particular emphasis was placed on mp-MRI–guided targeted biopsy and its integration with systematic sampling. Finally, reference lists from the retrieved articles were screened for additional relevant studies.

## 3. Prostate Biopsy Techniques: Historical and Current Perspectives

### 3.1. Transrectal and Transperineal Prostate Biopsy

As previously mentioned, prostate biopsy can be performed via systematic, targeted, and combined approaches, as well as different routes, such as transperineal and transrectal routes [8]. Historically, the first prostate biopsies were performed blindly, under finger guidance [9]. Over the past two decades, finger-guided prostate biopsies via the transrectal or transperineal route have been gradually abandoned in favor of TRUS prostate biopsies [9].

For many years, the standard practice concerned the performance of systematic biopsies without the prior imaging or localization of the suspicious lesions [10]. This approach required bilateral sampling, targeting both prostatic lobes and extending as far posteriorly and laterally as possible within the peripheral gland [11]. Interestingly, the number of cores taken from the prostate has been an area of conflict since the older six-core pattern was found to miss 10–30% of cancers [12]. In a systematic review conducted by Eichler et al., a minimum of 12 cores was recommended for systematic biopsies, while taking more than 12 cores was not associated with an increased detection of clinically significant cancer [12]. TRUS-guided biopsy still remains the main diagnostic procedure for prostate cancer in most clinical settings, despite evidence demonstrating its limitations in detecting csPCa and its propensity to overdiagnose indolent prostate cancer (isPCa) [13,14].

Nevertheless, despite these disadvantages, TRUS prostate biopsy continues to be widely performed under local anesthesia in most clinical settings due to its cost-effectiveness compared to alternative techniques [15]. Although the transrectal route is the most commonly used approach, current guidelines advocate using the transperineal approach because of its lower risk of infectious complications [16].

### 3.2. Comparing Transrectal and Transperineal Approaches

Current evidence suggests that the transrectal approach should be replaced by the transperineal approach, primarily due to its lower incidence of infection-related complications compared to the latter [16]. Specifically, in a meta-analysis of eight randomized studies, involving 1596 patients, infectious complications were significantly higher after transrectal biopsy (TRB) compared to TPB (risk ratio (RR): 2.48, 95% CI, 1.47–4.2) [17]. Furthermore, a systematic review of 165 studies by Bennet et al. estimated the infection rates after TPB and TRB at 0.1% and 0.9%, respectively [18]. Regarding overall cancer detection rates, there appears to be no significant difference between transrectal and tranperineal approaches [19].

However, a systematic review and meta-analysis comparing MRI-targeted biopsies via the transrectal and transperineal routes showed that TPB was associated with a higher csPCa detection rate (86% vs. 73%, respectively), especially for anterior tumors [20]. Another advantage of the transperineal approach is the minimal or nonexistent need for antibiotic prophylaxis. Interestingly, a meta-analysis of eight non-randomized controlled trials (RCTs) of patients undergoing TPB found no significant differences in post-biopsy infection rates (0.11% vs. 0.31%) or sepsis (0.13% vs. 0.09%) between those who received antibiotic prophylaxis and those who did not [21]. However, further well-designed RCTs are needed to clarify the role of antibiotic prophylaxis in the transperineal approach. A visual comparison of the transrectal vs. transperineal approaches is provided in Figure 1.

## 4. MRI/US Fusion Biopsy: A Diagnostic Advancement

### 4.1. Integration of Multiparametric MRI and Diagnostic Efficacy

The introduction of mp-MRI in everyday clinical practice has transformed the diagnostic pathway for prostate cancer. Suspicious lesions on mp-MRI are typically subjected to targeted biopsy along with standard systematic biopsies [22]. The process of MRI/ultrasound fusion-guided biopsy, combining both targeted and systematic biopsies, is presented in Figure 2.

According to the updated European Association of Urology (EAU) guidelines, the transperineal approach is recommended as the optimal technique, mainly due to its lower post-biopsy infectious rates [16]. However, whether an MRI-targeted TPB offers a diagnostic advantage over the transrectal route in detecting csPCa remains debatable [22].

Only few RCTs have compared detection rates between transrectal and transperineal approaches. Notably, in an RCT conducted by Hu et al., the detection rate of clinically significant cancer was similar between the two routes (53% transperineal vs. 50% transrectal, adjusted difference 2.0%; 95% CI −6.0, 10) [23]. Although the study included 658 patients with balanced randomization, its primary endpoint was the incidence of post-biopsy infections rather than cancer detection outcomes [23]. In a non-inferiority RCT by Ploussard et al., 270 MRI-positive biopsy-naïve patients were randomized 1:1 to either transrectal or transperineal MRI-targeted prostate biopsies [24]. The detection rates of significant PCa were similar (47.2% in TPB and 54.2% in the TRB approach, *p* = 0.6235) [24]. Regarding the per-lesion analysis, posterior lesions were better detected via the transrectal route (59.0% vs. 44.3%, *p* = 0.0443), while anterior lesions were more frequently detected via the transperineal route (40.6% vs. 26.5%, *p* = 0.2228) [24]. Finally, Mian et al. randomized 840 men to TRB or TPB and found similar overall cancer detection rates, 72.1% and 70.4%, and clinically significant cancer detection rates of 47.1% and 43.2% (OR: 1.17; 95% CI, 0.88–1.55), respectively [25]. Interestingly, MRI-targeted biopsies yielded clinically significant detection rates of 59% (TRB) and 62% (TPB) [25].

An earlier systematic review and meta-analysis by Tu et al. evaluated the diagnostic accuracy of fusion MRI/US TRB versus TRB in detecting csPCa [20]. Among patients with suspicious mp-MRI lesions, targeted biopsies via the tranperineal route demonstrated a higher detection rate (62.2%) compared to the TR route (41.3%) [20]. In the same study, when systematic and targeted biopsies were combined, the transperineal approach was related to an increased incidence (91.3%) of csPCa compared to the transrectal approach (72.2%) [20]. In a more recent meta-analysis conducted by Uleri et al., no statistically significant difference for MRI-targeted biopsies was found between the transrectal and transperineal approach [22]. More specifically, targeted MRI biopsy via the tranperineal route was associated with higher detection rates of csPCa anterior lesions (OR = 2.17, *p* < 0.001) and apical lesions (OR = 1.86, *p* = 0.01), while no statistically significant difference was found for posterior lesions [22]. Stratifying the results based on prostate imaging reporting and data system (PI-RADS) scores revealed that TPB was significantly more effective in diagnosing csPCa in PI-RADS 4 lesions (OR 1.57, *p* = 0.02), whereas this significant difference was not observed in PI-RADS 3 and 5 lesions [22].

In another meta-analysis that included 15 published studies, transperineal MRI/US fusion targeted biopsy achieved a higher detection rate of csPCa in the per-patient analysis (RR 1.33, 95% CI 1.09–1.63, *p* = 0.005), while a non-significant comparable difference was observed in the per-lesion analysis (RR 0.91, 95% CI 0.76–1.08, *p* = 0.28) [26]. Furthermore, Loy et al. conducted a systematic review and meta-analysis that included fourteen articles with 2002 patients. In the same analysis, although the transrectal approach performed better than the transperineal approach, sensitivity (TRB vs. TPB: 0.81 vs. 0.80) and specificity (TRB vs. TPB: 0.81 vs. 0.80) were considered equivalent between the two methods [27]. Finally, in a recently published meta-analysis by Zattoni et al., which included only RCTs (three studies), no significant difference was found for the detection of csPCa and insignificant prostate tumors [28]. A comprehensive summary of all available meta-analyses published in the literature, comparing different routes of fusion MRI/US prostate biopsies, is provided in Table 1.

The major disadvantage of these systematic analyses is that they include mainly prospective or retrospective studies, with great heterogeneity between studies, and thus, they are subject to the systematic errors of observational studies and present contradictory findings. Although meta-analyses present with contradictory findings concerning the detection rate of prostate cancer, the majority of these studies show better detection rates for the TPB approach. These contradictory results are mainly due to the retrospective design of the published studies, including patients with varying clinical characteristics. In addition, the analysis of these studies concerns either per-patient or per-lesion analysis, showing different results based on the primary endpoint analysis. It is well described that TPBs have better detection rates for anterior lesions. However, the majority of published studies do not distinguish the suspected lesions based on their location, and thus, the varying findings may due to different locations of the lesion. At present, there have only been a few RCTs published in the literature that compare the detection rate of fusion MRI/US prostate biopsies between the tranperineal and transrectal route. Further well-designed studies are needed to elucidate the optimal route for the detection of csPCa in the setting of the mp-MRI period.

### 4.2. Targeted Biopsy vs. Systematic Biopsy

In the era of mp-MRI for prostate evaluation, when a suspicious lesion is identified, the standard biopsy technique concerns a combination of targeted and systematic sampling [30]. As far as systematic biopsies are concerned, a minimum of twelve biopsies should be taken bilaterally, from the apex to the base, posteriorly and laterally, in the peripheral zone [16]. For suspicious mp-MRI lesions, targeted biopsies can be obtained via various methods, such as cognitive guidance, US/MRI fusion or direct in-bore guidance [16]. Although, no single method has demonstrated clear superiority, herein, the discussion focuses exclusively on targeted biopsies obtained via the MRI/US fusion approach. Interestingly, the majority of well-designed studies that compare cancer detection rates between targeted and systematic biopsies concern the transrectal approach. However, since both systematic and targeted biopsies do not differ between the transrectal and transperineal approach, these findings are considered similar for patients subjected to TPB [31].

The incorporation of targeted biopsies enhances the detection rate of clinically significant prostate cancer (the International Society of Urological Pathology, ISUP grade ≥2) while decreasing the detection rate of non-significant prostate cancer (ISUP grade 1) [16]. More specifically, according to the PRECISE trial, a multicenter, prospective randomized clinical trial that included 453 biopsy-naive patients, targeted biopsies (35% detection rate) were non-inferior (absolute difference 5%) for the detection of clinically significant prostate cancer compared to systematic biopsies (30% detection rate) [32]. Furthermore, in another multicenter, randomized, non-inferiority trial, clinically significant prostate cancer was found in 38% and 26% of MRI-targeted cores and systematic cores, respectively, presenting with an adjusted difference of 12% (95% CI, 4 to 20; *p* = 0.005) [33]. In a meta-analysis of 29 included studies, the detection rate of clinically significant prostate cancer was higher in the MRI-targeted biopsies compared to systematic biopsies (risk difference 11%, 95% CI 0–20, *p* = 0.05) [34]. As a result, the inclusion of MRI-targeted cores in the biopsy technique is considered necessary in order to ensure higher detection rates of clinically significant tumors.

Several RCTs have investigated the role of MRI-targeted biopsies in the identification of non-significant prostate tumors. MRI-targeted biopsy alone decreases the overdiagnosis of non-significant prostate tumors compared to systematic biopsies [16]. More specifically, according to the 4M trial, a prospective, multicenter study that included 626 biopsy-naïve patients, the MRI pathway detected non-significant prostate cancer in 14% of patients, while systematic biopsies detected insignificant PCa in 25% of the included patients [13]. In addition, the PRECISION trial, a multicenter, randomized trial, showed that the percentage of insignificant PCa was lower in the MRI-targeted biopsy group compared to the standard-biopsy group (adjusted difference, -13 percentage points; 95% CI, −19 to −7; *p* < 0.001) [33].

According to the EAU guidelines, the optimal approach for performing fusion MRI/US biopsies constitutes the combination of MRI-targeted and systematic biopsies [16]. Of note, the addition of targeted biopsies to systematic biopsies in biopsy-naive patients increases the number of clinically significant tumors and, more specifically, increases the detection of ISUP grade ≥2 and grade ≥3 PCa by 20% and 30%, respectively. Conversely, performing only targeted biopsies and avoiding systematic biopsies in biopsy-naive patients are related with lower detection rates of ISUP grade groups ≥2 and ≥3 by 16% and 18%, respectively [16].

## 5. Perilesional Biopsy in the Era of Fusion Imaging

As previously discussed, combined procedures, including sextant-based systematic biopsies along with three to five targeted biopsies for MRI suspicious lesions, comprise, at present, the optimal choice for prostate biopsies [35]. However, the exact allocation of neoplastic tissue related to the region of the suspected lesion and how systematic biopsy adds sensitivity to the combined biopsy are not clear [30]. A possible mechanism explained by van der Leest et al. demonstrated that some of the ‘‘systematic’’ cores that were found with prostate cancer were actually obtained from regions near the MRI lesion (perilesional) [13]. According to the last update of the EAU guidelines for prostate cancer, additional perilesional biopsies should be taken, avoiding standard systematic biopsies in MRI-negative prostate zones, in order to reduce the total number of cores obtained and increase the rate of csPCa found [16].

More specifically, according to Brisbane et al., 90% of systematic cores that found csPCa were obtained within a 10 mm radius of the MRI suspected lesion [36], whereas the number of csPCa was diminished as the distance from the ROI was increased [30]. On the contrary, a meta-analysis found a non-significant difference for the detection of cancers with ISUP grade group ≥2 between the MRI-targeted and perilesional biopsy approach and the standard combined procedure of MRI-targeted and systematic biopsies [37]. Interestingly, the distance encompassing 90% of csPCa around a MRI-suspected lesion appears to depend on the PI-RADS of the lesion, and thus, the scope of perilesional biopsies should be adjusted based on the PI-RADS score of the identified lesion. [30]. Further well-designed studies are needed to define the existence of a perilesional cancer-containing radius around the MRI lesion, which would alter the standard practice.

## 6. Complications and Safety Considerations

TPB has become a more widely utilized approach to prostate cancer diagnosis due to its lower infection rates compared with transrectal approaches [9]. However, complications may still arise, including acute urinary retention, pain, and procedural risks. These must be understood by both clinicians and patients before making decisions about prostate cancer diagnoses.

One of the primary advantages of TPB over TRB is its lower risk of infectious complications. Numerous studies have shown lower rates of sepsis and other infections when using TPB compared to TRB. A systematic review highlighted that TPB tends to have a low incidence of febrile complications, or near-zero incidence, due to its avoidance of inoculating the prostate with rectal flora, which poses significant risks for infections during transrectal procedures [38,39]. In a cohort study, it was noted that TPB had no readmission rates due to sepsis or serious infectious complications, further proving its safety profile [40].

While TPBs may reduce infection risks, they still pose other complications, most notably acute urinary retention. Reports have noted higher urinary retention rates with TPBs compared to TRBs [31,41]. One possible explanation could be differences in anatomical and procedural processes when accessing the prostate through the perineum; this may temporarily impact urinary function.

Pain management is another key component of the TPB experience. While studies indicate that the pain associated with these procedures can be significant, local anesthesia has proven successful at mitigating discomfort [42,43]. The incidence of pain ranges between 9.1% and 33.5% for this procedure, indicating that while most patients tolerate the procedure well, the anesthetic technique and choice can significantly alter pain outcomes [31,42]. This emphasizes the need for individualized patient care services.

Complications associated with biopsies may also stem from technical aspects. For instance, in patients with prior abdominoperineal resection, practitioners may encounter difficulties visualizing and positioning the prostate biopsy needle accurately, which could result in inadequate sampling or missed lesions [44,45]. These technical challenges underline the necessity for experienced practitioners and the use of advanced imaging guidance, such as MRI/US fusion, to increase accuracy [46,47].

In the majority of high-volume institutions that perform TPB, these procedures are performed under general anesthesia in order to reduce the perceived pain due to the exaggerated positioning of the patient required to place a stepper grid and carry out skin punctures that are taken during the biopsy procedure [48]. Interestingly, some surgeons prefer to perform pudendal nerve block postoperatively in order to reduce post-biopsy pain. However, there is a trend to perform TPBs under local anesthesia in order to reduce both cost and complications after general anesthesia. A prospective cohort study conducted by Hogan et al. demonstrated that transperineal biopsies under local anesthesia are well tolerated, with 95.2% of patients able to complete the procedure [48]. Of note, the maximum pain score was recorded during ultrasound probe insertion, and the firing of the biopsy gun was tolerable [48]. Furthermore, although post-biopsy acute urinary rates are similar between TPB and TRB, this complication has been related to various factors, such as an enlarged prostate volume and post-biopsy infection [49]. As a result, patients that are subjected to prostate biopsies should undergo a careful pre-operative investigation in order to predict the occurrence of various complications. However, at present, there are no clear guidelines in relation to pre-biopsy factors, local anesthesia postoperatively, and the number of cores required, and these factors vary widely depending on surgeon preference and may influence both post-biopsy perception and complication rates.

The learning curve associated with TPBs can play a significant role in complications. Just like any surgical procedure, experience is key for successful results. Multicenter evaluations have shown that as practitioners gain experience using the transperineal technique, complication rates tend to decline [50,51]. This emphasizes the significance of training and mentorship when adopting new biopsy techniques. Although the transperineal approach presents with various advantages, this approach has not been adopted by all institutes yet due to financial and technical reasons. Firstly, adaptation of the TPB approach was found to be difficult among men with low income and low-volume hospitals, while several disparities in the usage of this new technique have also grown for patients found in rural areas compared to urban centers and public versus private clinics [28]. In addition, transperineal procedures are usually performed under general anesthesia or after the performance of nerve blockage and thus require the assistance of anesthesiologists, especially those with experience in interventional radiology procedures [31]. Finally, for surgeons to gain proficiency in using these techniques, transperineal fusion MRI/US biopsies seem to necessitate a longer learning curve compared to the transrectal approach [52]. As a result, TPB is considered a costly and time-consuming procedure, requiring the use of general anesthesia and expensive equipment, explaining why it has not been adopted in every-day clinical practice.

Consideration must be given not only to immediate complications but also to long-term outcomes. Studies have revealed that while TPBs are linked with lower rates of acute complications, their long-term impacts on urinary function and sexual health vary greatly. For instance, repeat TPBs could increase the risk of long-term erectile dysfunction, though the extent of such risks remains a matter of investigation [53,54]. The transperineal approach also eliminates the risk of rectal bleeding, a potentially serious complication of TRBs, due to its avoidance of the rectal mucosa [55].

Furthermore, the psychological implications of prostate biopsies, regardless of their approach, must not be underestimated. Anxiety associated with the procedure and the possibility of a cancer diagnosis may significantly diminish patient outcomes and satisfaction. Providing pre-procedure counseling and support could reduce some adverse psychological effects [56,57].

## 7. MRI/US Fusion Biopsy in Active Surveillance

Active surveillance (AS) constitutes the optimal approach for the management of patients with non-significant prostate cancer (ISUP grade 1) with very-low-risk and low-risk prostate cancer (PCa). The primary aim is to minimize overtreatment and subsequent morbidity from radical prostatectomy [37] or radiation therapy and, at the same time, achieve the correct timing for curative treatment [58].

The inclusion criteria for AS are based on systematic biopsies and usually concern ISUP grade group 1, clinical stage cT1c or cT2a, PSA < 10 ng/mL, and PSA density (PSA-D) < 0.15 ng/mL/cc [59,60]. According to the EAU guidelines, in male candidates for AS based on systematic biopsy findings alone, a re-biopsy within six to twelve months, in order to confirm the initial findings, is considered necessary [16]. If the PCa diagnosis is made based on MRI-targeted biopsy alone, in order to avoid including patients with non-significant PCa, a confirmatory systematic biopsy should be performed [8,13,61].

However, men eligible for AS that undergo combined systematic and MRI-targeted biopsy do not require a confirmatory biopsy [16]. The misclassification of patients with non-significant disease based on the biopsy criteria of the first standard biopsy ranges from 20% to 30% of patients that undergo prostate biopsy [62]. Interestingly, upon adding MRI-targeted biopsy to systematic biopsy, the detection of ISUP grade group ≥2 cancer increased from 20% to 27% [63].

In another review, it was shown that combining targeted and systematic biopsies improves csPCa detection rates by 5 to 15% compared to targeted biopsies alone [64]. A meta-analyses of six studies conducted by Schoots et al. investigated the role of MRI/US fusion combined with targeted and systematic biopsies in men with low-risk disease opon TRUS biopsy. The findings revealed that 27% of the included patients were upgraded to csPCa using a combined approach compared to 17% and 20% of patients with targeted and systematic biopsy alone, respectively [63]. Interestingly, many candidates for AS that ultimately progressed within 1 year with more advanced disease may have had a more advanced disease initially that was misdiagnosed during the first TRUS biopsy [62].

As a result, although systematic biopsy constitutes the initial approach for patients under AS, a combined approach, including targeted and systematic biopsies, actually maximizes cancer detection rates and does not require the performance of further biopsies in the initial setting [65].

## 8. Future Perspectives

Fusion MRI/US TPB has emerged as a significant advancement in the diagnosis of prostate cancer, offering improved accuracy and patient outcomes compared to traditional methods. As technology continues to evolve, several promising developments are poised to further enhance the efficacy and safety of TPB in clinical practice.

One of the foremost advancements lies in the enhanced precision afforded by advanced imaging techniques. The integration of high-resolution imaging systems, such as micro-ultrasound and artificial intelligence (AI)-assisted MRI/US fusion, is anticipated to refine lesion targeting significantly. Further well-designed computer-aided diagnostic systems have been developed and validated to provide a better characterization of mpMRIs and improve prostate cancer diagnosis [66]. These technologies aim to reduce false negatives and improve the detection rates of csPCa, as indicated by Rodríguez Socarrás et al. (2020) [67].

Minimizing biopsy-related complications remains a critical focus in the evolution of TPB. The transperineal approach inherently reduces infection risks by avoiding the rectal mucosa. Ongoing refinements in anesthesia protocols and needle placement techniques are expected to lower the incidence of urinary retention and procedural pain, enhancing patient comfort and safety [68].

Few published studies have shown that higher anxiety levels during or after biopsy procedures are related to increased pain perception and a longer duration of the procedure, resulting in lower patient satisfaction [69]. Upon the diagnosis of prostate cancer, symptoms of anxiety and depression can be found in 30,3% and 20,3% of patients, respectively [69]. After the completion of a biopsy procedure, the incidence of events, such as sepsis, hematuria or hemospermia, can also significantly increase anxiety levels [69]. Identifying effective techniques to manage anxiety is considered crucial for improving the patient experience, while preoperative education and anxiety management are able to decrease both postoperative anxiety levels and the perception of complications [69].

The incorporation of artificial intelligence in biopsy planning and analysis represents another frontier in TPB. AI tools are projected to improve the identification of high-risk lesions on MRI and streamline the fusion process between MRI and ultrasound images. These advancements may increase diagnostic efficiency and reduce inter-observer variability in lesion assessments, contributing to more consistent and accurate diagnoses [29].

Technological improvements have also facilitated the development of outpatient TPB practices. The reduced necessity for antibiotics due to lower infection risks and the feasibility of performing the procedure under local anesthesia make TPB a more accessible and patient-friendly option. This shift towards outpatient settings could enhance patient throughput and reduce healthcare costs [70].

Integrating TPB with risk stratification models enhances its utility in personalized medicine. The use of MRI-based scoring systems, such as the PI-RADS and the PRECISE criteria, enhances the predictive value of biopsies. This integration supports individualized decision making for active surveillance and treatment planning, tailoring interventions to patient-specific risk profiles [71].

The expansion of research through clinical trials is expected to solidify the role of TPB in prostate cancer diagnostics. Ongoing randomized controlled trials, such as PERFECT and TRANSLATE, are anticipated to provide robust evidence comparing the efficacy of TPB and traditional TRB approaches. The outcomes of these studies may establish TPB as the gold standard in prostate cancer diagnosis [24,36,72].

Finally, an exploration of novel applications of TPB is underway. The technique’s ability to identify anterior and apical lesions has been demonstrated, and future research may focus on leveraging MRI/US fusion-guided TPB to deliver targeted therapies and focal treatments directly to the prostate. This could open avenues for minimally invasive, localized treatment options for patients [73].

Restricted access to specific transperineal probe and biopsy equipment, functional costs and difficulties in the management of peri-operative pain comprise the major barriers to the adaptation of the transperineal approach as the optimal approach [74]. Nonetheless, multiple studies have shown that with a well-distributed regional (pudendal, perineal, and periprostatic) block using local anesthesia, the procedure is tolerable, with rarely a need for sedation [75,76]. Furthermore, although the initial cost of transperineal equipment is considered high, long-term costs per procedure for transperineal and transrectal approaches are equal with future cost reduction due to the minimization of septic events or hospital re-admissions [74,77]. Interestingly, the exposure of residents to tranperineal procedures is related to increased intent to use TPB after training, while even a limited degree of exposure to this novel biopsy technique is associated with the increased utilization of this approach in their future career [77]. Of note, institutions that lack both the experience and financial resources to provide training for TPB may consider allowing residents to undergo short visits and rotations at other institutions in order to gain exposure to these approaches and thus be more likely to use it in practice.

## 9. Conclusions

The integration of fusion MRI/US TPB into the diagnostic approach for prostate cancer represents a significant advancement over traditional methods. This technique enhances the detection of clinically significant prostate cancers by combining the precision of mp-MRI with the advantages of the transperineal route, notably reducing infection risks and improving access to anterior and apical lesions. The evidence underscores that combining targeted MRI/US fusion biopsies with systematic sampling maximizes the diagnostic yield while minimizing the overdiagnosis of clinically insignificant tumors, thus refining patient selection for appropriate management strategies.

Compared with the well-established transrectal approach, TPB demonstrates higher detection rates for suspicious lesions in the anterior prostate and lower post-biopsy infection rates. However, published studies have not consistently identified a significant difference in overall or clinically significant prostate cancer detection between these two approaches. Finally, although TPB can be costly and requires specialized expertise and equipment, targeted training and investment may reduce long-term expenses by lowering hospitalizations, antibiotic usage, and related costs. Further well-designed RCTs are needed to elucidate the primary role of TPB in every-day clinical practice.

## Figures and Tables

**Figure 1 jcm-14-00453-f001:**
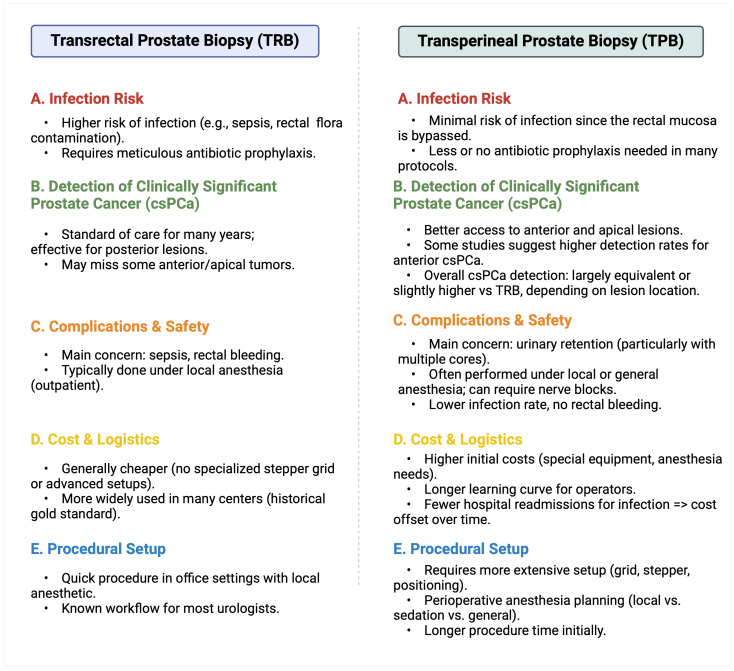
Transperineal vs. transrectal prostate biopsy: a comparative overview. Created in BioRender. Kaltsas, A. (2025), https://BioRender.com/x02m091 (accessed on 6 January 2025).

**Figure 2 jcm-14-00453-f002:**
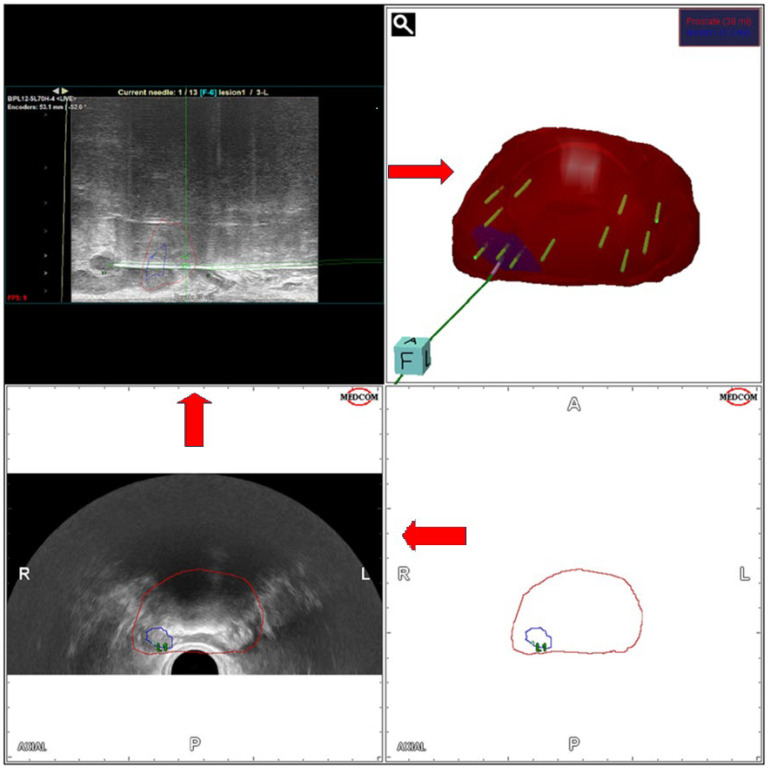
The process of MRI/ultrasound fusion-guided biopsy, combining targeted and systematic biopsies. Initially, the first step is planning the prostate boarders and identifying the location of the suspected lesion. One must undertake both targeted biopsies based on the location of the suspected lesion, and systematic biopsies based on the initial planning and prostate characteristics. The number of the undertaken targeted biopsies should be based on the size of the suspected lesion. Systematic biopsies should be taken bilaterally, from the base to the apex of the prostate, six in each lobe. Red marked area: prostate, Blue marked area: lesion, Green arrow: needle biopsy, A: Anterior, P: Posterior, R: Right, L: Left.

**Table 1 jcm-14-00453-t001:** Summary of meta-analyses comparing transperineal and transrectal fusion MRI/US prostate biopsy for detection of clinically significant prostate cancer.

Study	Year	Study Type	Patients (n)	Detection Rate of csPCa(TPB vs. TRB)	Key Findings
Wu et al. [26]	2024	15 Studies5 Prospective 9 Retrospective 1 RCT	8826 pts	Per-patient analysis: (one cohort (RR 1.33 [95% CI 1.09, 1.63], *p* = 0.005)) Per-lesion analysis (RR 0.91 [95% CI 0.76, 1.08], *p* = 0.28)	TPB was associated with higher csPCa detection rates in cases of anterior lesions: per lesion: RR = 1.52, 95% CI 1.04, 2.23) per patient (RR 2.55, 95% CI 1.56, 4.16).
Zattoni et al. [28]	2024	3 RCTs	1599 pts	OR = 0.9, 95% CI: 0.7–1.1	ISUP 1 prostate cancer (PCa; OR 1.1, 95% CI: 0.8–1.4) detection.
Uleri et al. [22]	2023	11 Studies 9 Retrospective 1 Prospective 1 RCT	3522 TRB vs. 5140 TPB	OR = 1.11, 95% CI, 0.98–1.25, *p* = 0.1	TPB was associated with higher csPCa detection rates in cases of anterior (OR = 2.17, 95% CI 1.46–3.22) and apical (OR 1.86, 95% CI 1.14–3.03) lesions. TPB was associated with higher csPCa detection rates (OR 1.57, 95% CI 1.07–2.29) in PI-RADS 4 lesions.
Rai et al. [29]	2021	2 Retrospective	350 pts	Per-patient analysis: RR 1.28, 95% CI, 1.03–1.60, *p* = 0.03 Per-lesion analysis: 65.7% and 75.5% (*p* = 0.40)	Per lesion: MRI-TPB was associated with better detection of anterior csPCa lesions (RR: 2.46, 95% CI 1.22–4.98). Per lesion: MRI-TPB and MRI-TRB overall cancer detection rates were 75% and 81.6% (*p* = 0.53).
Tu et al. [20]	2019	4 Studies 3 Prospective1 Retrospective	315 TRB vs. 328 TPB	OR= 2.37, 95% CI, 1.71–3.26	TRB missed more csPCa lesions located at the anterior zone of the prostate (20 vs. 3). TPB performed better than the TRB route in MRI-targeted biopsy, especially in detecting csPCa located at the anterior prostate.

Abbreviations: PCa: prostate cancer; csPCa: clinical significant prostate cancer; TPB: transperineal biopsy; TRB: transrectal biopsy; ISUP: International Society of Urological Pathology; RCT: randomized controlled trial; MRI: magnetic resonance imaging; OR: odds ratio, RR: relative risk; CI: confidence interval; pts: patients.

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
