# Peer review of "Fusion MRI/Ultrasound-Guided Transperineal Biopsy: A Game Changer in Prostate Cancer Diagnosis"

_jcm, 2025, doi:10.3390/jcm14020453_

Round 1
Reviewer 1 Report
Comments and Suggestions for Authors
The paper is well structured, the English needs to be revised almost completely in a more discursive manner, redundant sentences about the purpose of the paper need to be removed, and a paragraph added about why if it is so advantageous compared to the tarnsrectal approach it is still not widely used (costs, learning curve, Rm availability...).

Comments on the Quality of English Language
the use of more discursive English is more suited to the narrative approach given to the article, which is why a revision of the text should be carried out and especially the elimination of unnecessary redundant sentences too often repeated throughout the text
Author Response
REVIEWER 1
Comments to the Author
The article was revised and English language was corrected extensively and improved by the authors, as the reviewers indicated.
ABSTRACT
- … prostate cancer undergo this diagnostic modality. Mp-MRI presents with the ability…
Rephrase, eliminate “with”
The phrase “with” was eliminated as you proposed.
- … Finally, transperineal Fusion MRI/US biopsy can be useful for the repeat biopsy for patients with initial negative biopsy
Rephrase
Thank you for your constructive remark. The sentence was replaced by the following phrase: “Finally, transperineal Fusion MRI/US biopsy can be valuable for repeat biopsies in patients who had an initial negative biopsy”
MAIN TEXT
- 3.1 Until now, there are only few RCTs published in the literature to compare the detection rate of Fusion MRI/US prostate biopsies between the tranperineal and transrectal route..
More emphasis should be placed on individual RCTs in addition to the good summary table
mentioned previously
Thank you for your valuable comment. The following text was added in the Integration of Multiparametric MRI and Diagnostic Efficacy section as you proposed:
“Only few RCTs have compared detection rates between transrectal and transperineal approaches. Notably, in a RCT conducted by Hu et al, the detection rate of clinically significant cancer was similar between two routes (53% transperineal vs 50% transrectal, adjusted difference 2.0%; 95% CI –6.0, 10) [23]. Although that study included 658 patients with balanced randomization, its primary endpoint was the incidence of post-biopsy infections rather than cancer detection outcomes [23]. In a non-inferiority RCT by Ploussard et al., 270 MRI-positive biopsy-naïve were randomized 1:1 to either transrectal or transperineal MRI-targeted prostate biopsy [24]. The detection rates of significant PCa were similar (47.2% in TPB and 54.2% in TRB approach, p = 0.6235) [24]. On a per-lesion analysis, posterior lesions were better detected via transrectal route (59.0% vs 44.3%, p = 0.0443), while anterior lesions were more frequently detected via transperineal route (40.6% vs 26.5%, p = 0.2228) [24]. Finally, Mian et al. randomized 840 men to TRB or TPB and found similar overall cancer detection rates: 72.1% and 70.4%, and clinically significant cancer detection rates of 47.1% and 43.2%, (OR: 1.17; 95% CI, 0.88-1.55), respectively [25]. Interestingly, MRI-targeted biopsies yielded clinically significant detection rates of 59% (TRB) and 62% (TPB) [25].”
- ..3.2… Interestingly, the majority of well-designed studies that compare the cancer detection rates between targeted and systematic biopsies concern TRUS approach. However, since both systematic and targeted biopsies do not differ between TRUS and TP approach, these findings are considered sim- ilar for patients subjected to TP biopsy.
is there any literature to technically superimpose the two sampling methods or is it an
assumption of the author? explain more clearly
Thank you for your interesting question. This is an assumption made by the authors, but also is based on evidence of the literature. In a study conducted by Gilberto et al. comparing transperineal to transrectal approach, he mentioned that “the main difference between the two approaches is the access route, but the biopsy is performed by the same technique; true cut needle and US guided with MRI software fusion”. Consequently, the reference was added in the text to superimpose the two sampling methods.
- … The learning curve associated with transperineal biopsies can play a significant role in complications. Just like any surgical procedure, experience is key for successful results. Multicenter evaluations have shown that as practitioners gain experience using the trans- perineal technique, complication rates tend to decline [44,45]. This emphasizes the signif- icance of training and mentorship when adopting new biopsy techniques.
if technically the procedure is superior, why isn't it already being adopted by everyone globally? exploring the interesting subject of costs and the learning curve would make the text more attractive
Thank you for your interesting comment. As already mentioned, the transperineal approach seems to be superior compared to transrectal approach in terms of cancer detection rate, especially for anterior tumors and infection rate. However, as you commented, this approach has not been adopted yet in all countries, due to various difficulties. The following sentence was added in the Complications and Safety Considerations section:
“Although the transperineal approach presents with various advantages, this approach has not been adopted by all institutes yet, due to financial and technical reasons. Firstly, adaptation of the TP approach was found difficult among men with low income and low-volume hospitals, while also several disparities in the usage of this new technique have grown for patients found in rural areas compared to urban centers and public versus private clinics [52]. In addition, transperineal procedures are usually performed under general anesthesia or after performance of nerve blockage, and thus, recquires the interference of anesthesiologists, and especially, with experience in interventional radiology procedures [31]. Finally, transperineal Fusion MRI/US biopsy seem to have longer learning curve compared to transrectal approach in order to gain proficiency [53]. As a result, TPB is considered a costly and time-consuming procedure, requiring the use of general anesthesia and expensive equipment, explain why it has not been adopted in every-day clinical practice.”
- … While transperineal biopsy provides many advantages, particularly regarding infec- 276
tion risk, it is crucial that practitioners remain cognizant of potential complications asso- 277
ciated with the procedure. Acute urinary retention, pain management challenges, and 278
technical hurdles must all be considered. Ongoing research and clinical experience will 279
continue to refine the techniques and protocols involved with transperineal biopsies, thus 280
improving patient outcomes and satisfaction.
redundant, already mentioned in the text
Thank you for your valuable remark. The following phrases were erased as you proposed.
7.
8.… Looking ahead, the continued evolution of this diagnostic modality is poised to further improve patient outcomes. The incorporation of advanced imaging technologies and artificial intelligence stands to enhance lesion detection and targeting accuracy. Developments in outpatient transperineal biopsy practices and integration with risk stratification models support a more patient-centered and efficient healthcare delivery. Ongoing clinical trials are expected to provide robust evidence to establish fusion MRI/US trans- perineal biopsy as the gold standard in prostate cancer diagnostics, paving the way for more personalized and effective treatment approaches.
merge the two paragraphs future perspectives and conclusions by shortening the text, which
is too repetitive. Instead, I would add a paragraph on what can be done to further increase the diffusivity of the transperineal approach and its teaching (learning curve), as the benefits are clearly demonstrated.
Thank you for your constructive note. As you proposed, the text you indicated was erased.
Also, a paragraph was added describing how transperineal approach can be more diffusive in the Future Prespective section; “As already mentioned, restricted access to specific transperineal probe and biopsy equipment, along with functional costs and difficulties in the management of peri-operative pain comprise the major barriers to adaptation of transperineal approach as an optimal approach [75]. Nonetheless, multiple studies have shown that with a well-distributed regional (pudendal, perineal, and periprostatic) block using local anesthesia the procedure is tolerable with rare need for sedation [76,77]. Furthermore, although the initial cost of transperineal equipment is considered high, long-term costs per procedure between transperineal and transrectal approaches are equal with future cost reduction due to minimization of septic events or hospital re-admissions [75,78]. Interestingly, exposure of residents with the tranperineal procedures is related to increased intent to use the transperineal biopsy after training, while even a limited degree of exposure to this novel biopsy technique is associated with increased utilization of this approach in their future career [78]. Of note, institutions that lack both the experience and financial resources to provide training for TP-Bx may consider to allow short visits and rotations at other institutions in order to gain exposure to these approaches and thus be more likely to use it in practice.”
Reviewer 2 Report
Comments and Suggestions for Authors
Dear Authors,
i reviewed with interest your review.
I have minor comments to report
1. Please re-formulate the abstract including method, results and conclusion
2. Please describe the study methods in a dedicated chapter
3. Please comment on the clinical implication of the study findings
4. Please include the following paper when discussing on new technology in MRI and prostate biopsy (doi: 10.1007/s00345-022-04275-x)
Author Response
Comments to the Author
i reviewed with interest your review.
I have minor comments to report
- Please re-formulate the abstract including method, results and conclusion
Thank you for your constructive comment. The Abstract was re-formulated as you proposed. The following sentences were added in Backround/Objectives section: “The aim of this comprehensive review is to highlight the role of both standard and targeted MRI/US fusion transperineal biopsy (TPB) in the diagnostic approach of prostate cancer cases, to report its diagnostic efficacy and complication rates and to suggest promising usage of MRI/US fusion TPB in the future”, in Methods section: “A comprehensive review of existing literature, including systematic reviews, meta-analyses, and clinical guidelines, was conducted to compare the efficacy and safety of transperineal and-transrectal approaches in prostate cancer detection. Special emphasis was placed on mp-MRI- guided targeted biopsy and its combination with systematic sampling.” and in Conclusion section: “MRI/US fusion-guided TPB represents a significant advancement in prostate cancer diagnostics, combining improved precision with reduced infection risks. Future research should further refine this approach and explore its integration with emerging technologies like artificial intelligence for enhanced lesion targeting and diagnostic accuracy.”
- Please describe the study methods in a dedicated chapter
Thank you for your remark. The study Methods was added, along with the following text:
A comprehensive literature search was conducted through January 2024 to iden-tify relevant articles. The PubMed database served as the primary source, and only English-language publications were included. The following search terms were used: “Transperineal Fusion MRI/US prostate biopsy,” “Fusion MRI/US prostate biopsy,” and “Transperineal prostate biopsy.” Searches focused on prostate biopsy techniques, MRI/US fusion biopsies, perilesional biopsy, complications, and safety considerations. Systematic reviews, meta-analyses, and clinical guidelines that compared the efficacy and safety of transperineal and transrectal approaches in prostate cancer detection were selected. Particular emphasis was placed on mp-MRI–guided targeted biopsy and its integration with systematic sampling. Finally, reference lists from the retrieved articles were screened for additional relevant studies.
- Please comment on the clinical implication of the study findings
Thank you for your valuable addition. The following text was added in the Conclusion section, concerning clinical implication of study findings; “Compared with the well-established transrectal approach, TPB demonstrates higher detection rates for suspicious lesions in the anterior prostate and lower post-biopsy infection rates. However, published studies have not consistently identified a significant difference in overall or clinically significant prostate cancer detection between these two approaches. Finally, although TPB can be costly and requires specialized expertise and equipment, targeted training and in-vestment may reduce long-term expenses by lowering hospitalizations, antibiotic us-age, and related costs.”
- Please include the following paper when discussing on new technology in MRI and prostate biopsy (doi: 10.1007/s00345-022-04275-x)
The following paper along with the above text was included, as you proposed: “Further well-designed computer-aided diagnostic systems have been developed and validated to provide better characterization of mpMRIs and improve prostate cancer diagnosis [67].”
Reviewer 3 Report
Comments and Suggestions for Authors
The article provides an extensive overview of transperineal prostate biopsy, its diagnostic accuracy, and its advantages over the traditional transrectal approach. The integration of multiparametric MRI with ultrasound fusion is clearly explained, making the topic accessible to both clinicians and researchers. The inclusion of meta-analyses and systematic reviews adds credibility to the findings. The discussion of studies comparing diagnostic outcomes, safety profiles, and infection rates between transperineal and transrectal biopsies is detailed and evidence-based. The article effectively addresses future developments, including the potential of artificial intelligence and advancements in imaging technologies. It highlights the ongoing clinical trials and their implications for establishing transperineal biopsy as the gold standard. The structure of the paper is logical, moving from the introduction and background through techniques, comparisons, complications, and future directions. This makes it easy to follow.
Recommendations for Improvement:
1. Some sections, especially those comparing detection rates between the transperineal and transrectal approaches, report conflicting results across studies. For instance, detection rates of clinically significant prostate cancer differ among meta-analyses. A synthesis or critique of these discrepancies would strengthen the discussion. Summarize the limitations of existing studies to guide readers in understanding these inconsistencies.
2. While the article mentions acute urinary retention and pain management, it could benefit from a more detailed exploration of strategies to minimize these complications, particularly in outpatient settings. Discuss the role of advancements in anesthesia and patient preparation protocols in more depth
3. Including flowcharts or diagrams to represent the process of MRI/ultrasound fusion-guided biopsy would aid understanding. Graphical summaries of comparative data on detection rates and complication incidences from different studies could make the findings more digestible.
4. Discuss barriers to the widespread adoption of transperineal biopsies, such as the required expertise, costs, and training. Address how these barriers might be mitigated. Include a brief mention of cost-effectiveness comparisons to support decision-making by healthcare providers.
5. The psychological impact and patient-reported outcomes (e.g., anxiety, satisfaction, and perceived pain) could be elaborated on. This would add a patient-centric view to the discussion.
The article is a valuable contribution to the field of prostate cancer diagnostics, providing a thorough analysis of fusion MRI/ultrasound-guided transperineal biopsy. Addressing the recommendations above would enhance its clarity, applicability, and overall impact. Specifically, integrating visuals, synthesizing conflicting evidence, and broadening the discussion on patient-centric aspects would elevate its utility for both clinical and research purposes.
Author Response
REVIEWER 3
Comments to the Author
- Some sections, especially those comparing detection rates between the transperineal and transrectal approaches, report conflicting results across studies. For instance, detection rates of clinically significant prostate cancer differ among meta-analyses. A synthesis or critique of these discrepancies would strengthen the discussion. Summarize the limitations of existing studies to guide readers in understanding these inconsistencies.
Thank you for your valuable remark. As already mentioned in this section, these systematic analyses present with contradictory findings, since they mainly include retrospective studies and are subsequent to systematic errors. However, the following text was added in order to elucidate these discrepancies in the MRI/US Fusion Biopsy: A Diagnostic Advancement section: “Although meta-analyses present with contradictory findings concerning the detection rate of prostate cancer, the majority of these studies show better detection rates for the TPB approach. These contradictory results are mainly due to retrospective design of the published studies, including patients with varying clinical characteristics. In addition, analysis of these studies concerns either per-patient or per-lesion analysis, showing different results based on the primary endpoint analysis. It is well-described that TPBs have better detection rates for anterior lesions. However, the majority of published studies do not distinguish the suspected lesions based on their location and, thus varying findings may due to different location of the lesion.”
- While the article mentions acute urinary retention and pain management, it could benefit from a more detailed exploration of strategies to minimize these complications, particularly in outpatient settings. Discuss the role of advancements in anesthesia and patient preparation protocols in more depth
Thank you for your valuable comment. The following sentence was added in the “Complications and Safety Considerations” section: In the majority of high-volume institutions that perform TPBs, these procedures are performed under general anesthesia, in order to reduce perceived pain due to the exaggerated positioning of the patient required to place a stepper, grid and skin punctures that are taken during biopsy procedure [48]. Interestingly, some surgeons prefer to perform pudendal nerve block postoperatively, in order to reduce post-biopsy pain. However, there is a trend to perform TPBs under local anesthesia, in order to reduce both cost and complications after general anesthesia. A prospective cohort study conducted by Hogan et al., demonstrated that TPBs under local anesthesia are well tolerated with 95.2% of patients able to complete the procedure [48]. Of note, the maximum pain score was recorded during ultrasound probe insertion, while the firing of the biopsy gun was tolerable [48]. Furthermore, although post-biopsy acute urinary rates are similar between TPB and TRB, this complication has been related with various factors, such as enlarged prostate volume and post-biopsy infection [49]. As a result, patients that are subsequent to prostate biopsies need careful pre-operative investigation in order to predict the occurrence of various complications. However, until now, there are no clear guidelines in relation to pre-biopsy factors, local anesthesia postoperatively, or the number of cores required, and these factors vary widely depending on surgeon preference and may influence both bost-biopsy perception and complication rates.”
- Including flowcharts or diagrams to represent the process of MRI/ultrasound fusion-guided biopsy would aid understanding. Graphical summaries of comparative data on detection rates and complication incidences from different studies could make the findings more digestible.
Thank you for your interesting suggestion. Figure 2 along with the description: “Figure 2: The process of MRI/ultrasound fusion-guided biopsy, compining targeted and systematic biopsies. Initially, the first step is planning the prostate boarders and identifying the location of the suspected lesion. Undertake both targeted biopsies, based on the location of the suspected lesion, and systematic biopsies, based on the intial planning and prostate characteristics. The number of the undertaken targeted biopsies should be based on the seize of the suspected lesion. Systematic biopsies should be taken bilaterally, from base to the apex of the prostate, six in each lobe.” was added in the text in order to represent the process of MRI/ultrasound Fusion-guided in order to aid understanding as you proposed. The Figure 1, along with the following phrase was added; “The process of MRI/ultrasound fusion-guided biopsy, undertaking both targeted and systematic biopsies, is presented in Figure 1.”, to aid understanding of the procedure as you proposed.
- Discuss barriers to the widespread adoption of transperineal biopsies, such as the required expertise, costs, and training. Address how these barriers might be mitigated. Include a brief mention of cost-effectiveness comparisons to support decision-making by healthcare providers.
Thank you for your interesting comment. Barriers to the widespread adoption of transperineal biopsies are being explained in the added text: “Although … clinical practice.”
The following sentences were added in the Future Perspectives section in order to explain how to mitigate these barriers, as you proposed: “As already mentioned, restricted access to specific transperineal probe and biopsy equipment, along with functional costs and difficulties in the management of pe-ri-operative pain comprise the major barriers to adaptation of transperineal approach as an optimal approach [75]. Nonetheless, multiple studies have shown that with a well-distributed regional (pudendal, perineal, and periprostatic) block using local an-esthesia the procedure is tolerable with rare need for sedation [76,77]. Furthermore, although the initial cost of transperineal equipment is considered high, long-term costs per procedure between transperineal and transrectal approaches are equal with future cost reduction due to minimization of septic events or hospital re-admissions [75,78]. Interestingly, exposure of residents with the tranperineal procedures is related to in-creased intent to use the TPB after training, while even a limited de-gree of exposure to this novel biopsy technique is aasociated with increased utilization of this approach in their future career [78]. Of note, institutions that lack both the ex-perience and financial resources to provide training for TP-Bx may consider to allow short visits and rotations at other institutions in order to gain exposure to these ap-proaches and thus be more likely to use it in practice.”
- The psychological impact and patient-reported outcomes (e.g., anxiety, satisfaction, and perceived pain) could be elaborated on. This would add a patient-centric view to the discussion.
Thank you for your valuable remark. The following sentences were added in the “Future Perspectives” section: Few published studies proved that higher anxiety levels during or after biopsies procedures are related with increased pain perception, longer duration of the proce-dure, resulting in lower patient satisfaction [70]. At diagnosis of prostate cancer, symptoms of anxiety and depression can be found in 30,3% and 20,3% of patients, re-spectively [70]. After the completion of biopsy procedure, incidence of events, such as sepsis, hematuria or hemospermia, can also significantly increase anxiety levels [70]. Identifying effective techniques to manage anxiety is considered crucial for improving the patient experience, while preoperative education and anxiety management are able to decrease both post-operative anxiety levels and perception of complications [70]
The article is a valuable contribution to the field of prostate cancer diagnostics, providing a thorough analysis of fusion MRI/ultrasound-guided transperineal biopsy. Addressing the recommendations above would enhance its clarity, applicability, and overall impact. Specifically, integrating visuals, synthesizing conflicting evidence, and broadening the discussion on patient-centric aspects would elevate its utility for both clinical and research purposes.
Thank you for your promising and delightful comments. We hope that integrating visual content, patient-centric aspects and reviewers interesting suggestions will increase the impact of the paper.